

# *Muvi schmallenbergi* gen. nov., sp. nov. (Crustacea, Tanaidacea) from the southeast Australian coast, with comments on the distribution and habitat preferences of Chondropodinae

Piotr Jóźwiak and Magdalena Błażewicz

Department of Invertebrate Zoology and Hydrobiology, Faculty of Biology and Environmental Protection, University of Łódź, Łódź, Poland

## ABSTRACT

Based on material collected from the shelf off southeast Australia (offshore of Portland), a new genus and new species, *Muvi schmallenbergi* gen. nov., sp. nov., of the tanaidacean family Metapseudidae, is described. *Muvi* is distinguishable from other genera within the subfamily Chondropodinae by having equally long antennular flagella. It also differs from other Chondropodinae by a combination of characters such as eyelobes with a group of visual elements, rostrum with smooth lateral edges, pereonites with lateral processes and pleotelson lacking lateral process, antennule article-1 with a single apophysis, maxillule inner lobe well-developed, labial palp bearing three distal setae, cheliped exopod well-developed and setose, pereopod-1 coxa with distinct apophysis, pleopods in five pairs, and uropod basis without apophysis. The genus *Deidamiapseudes* Sganga & Roccatagliata, 2016 is moved from Chondropodinae (Metapseudidae) to Apseudoidea *incertae sedis*. An identification key for the genera within Chondropodinae is given, and their distribution is discussed.

## INTRODUCTION

Tanaidacea, small benthic peracarid crustaceans, represent a poorly recognized component of marine ecosystems. Until the end of millennium, the Australian Tanaidacea were known only from few taxonomic publications (*Haswell, 1882a*, *1882b*; *Haswell, 1885*; *Whitelegge, 1901*; *Boesch, 1973*; *Băcescu, 1981*; *Sieg, 1993*; *Edgar, 1997*). The turning point was the publication of the results from several surveys dedicated to the tanaidacean fauna on the Australian shelf. For example, *Bamber (2005)* described 24 new species from Esperance Bay, *Guţu (2006a)* reported 13 new species from tropical zones of Australia, *Edgar (2008)* reported 12 new species of the family Tanaididae in Tasmanian waters, and *Błażewicz-Paszkowycz & Bamber (2012)* described 42 new species from the Bass Strait. As a result, the number of the tanaidaceans known to occur along the Australian coast increased from 22 to 209 (as summarized by *Bamber, 2008*; *Edgar, 2008*;

Corresponding author
Piotr Jóźwiak,
piotr.jozwiak@biol.uni.lodz.pl

*Błażewicz-Paszkowycz & Bamber, 2009*; *Błażewicz-Paszkowycz & Zemko, 2009*; *Stępień & Błażewicz-Paszkowycz, 2009a*, *2009b*; *Błażewicz-Paszkowycz & Bamber, 2012*; *Edgar, 2012*; *Jóźwiak & Jakiel, 2012*; *Bamber, 2013*; *Bamber & Błażewicz-Paszkowycz, 2013*; *Stępień & Błażewicz-Paszkowycz, 2013*; *Gellert & Błażewicz, 2018*). This demonstrates the high diversity of these small and poorly recognized peracarids along the Australian coast (*Bamber & Błażewicz-Paszkowycz, 2013*), with a high level of endemism (*Błażewicz-Paszkowycz & Bamber, 2012*; *Błażewicz-Paszkowycz, Bamber & Anderson, 2012*). However, the total number of Tanaidacea living in the Australian waters is still unknown (*Stępień, Pabis & Błażewicz, 2018*). Based on studies on the West Australian coast, *Poore et al. (2015)* reported almost 200 tanaidacean species, pointing them out as the most abundant taxon in terms of both individuals and species. This number, although double the previously known number of species from Australian waters, probably represents only a fraction of the tanaidacean fauna.

The metapseudid subfamily Chondropodinae is currently represented by 29 species classified to nine genera (*WoRMS, 2019*) distributed in tropical to temperate waters, *e.g.*, the Adriatic Sea, along the Brazilian coast, the Gulf of Guinea, Mauritania, the Gulf of Mexico, the Caribbean Sea, and the Coast of Malaysia (*Guțu, 1984*, *1996*, *2002*; *Bamber & Sheader, 2005*; *Guțu, 2006a*, *2014*; *Jakiel et al., 2015*). Two species of Australian Chondropodinae are currently known: *Julmarichardia gutui Ritger & Heard, 2007*, found along the Northwest Australian coast (*Ritger & Heard, 2007*), and *Bamberus jinigudirus Stępień & Błażewicz-Paszkowycz, 2013*, collected from Ningaloo coral reefs (*Stępień & Błażewicz-Paszkowycz, 2013*). Described herein, *Muvi schmallenbergi* sp. nov. is the third Chondropodinae species recorded from Australia.

## MATERIALS AND METHODS

The analyzed sample was taken during the SLOPE campaign off Portland, Victoria, Australia at the depth of 49.5 m, using a Smith–McIntyre grab. The sample was preserved in formaldehyde, identified, and fixed in 70% ethanol. Images of body habitus were taken with a Leica M125 stereomicroscope combined with a DFC295 camera and the LAS V4.5 software. Appendages were dissected in a glycerin solution using chemically sharpened tungsten needles, mounted in glycerin on slides, and sealed with nail varnish. Drawings were made using a Nikon Eclipse 50*i* microscope combined with a camera lucida, redrawn with China ink, and finally edited and cleared with Corel PHOTO-PAINT X7. The body length-to-width ratio was calculated using measurements from the tip of the carapace to the end of the pleotelson and of the widest part of the carapace; the length and width of the articles were measured along their central axes. The general morphological terminology followed that proposed by *Błażewicz-Paszkowycz, Bamber & Jóźwiak (2013)*. To simplify species descriptions, the expression '*N*x' replaces '*N* times as long as' and '*N* L:W' replaces '*N* times longer than wide'. The type material was deposited in the Melbourne Museum (NMV, Australia). Distribution maps were generated using the freeware QGIS.

The electronic version of this article in Portable Document Format (PDF) will represent a published work according to the International Commission on Zoological Nomenclature

(ICZN), and hence the new names contained in the electronic version are effectively published under that Code from the electronic edition alone. This published work and the nomenclatural acts it contains have been registered in ZooBank, the online registration system for the ICZN. The ZooBank LSIDs (Life Science Identifiers) can be resolved and the associated information viewed through any standard web browser by appending the LSID to the prefix http://zoobank.org/. The LSID for this publication is: urn:lsid: zoobank.org:pub:E516068D-B9FC-4267-BC3C-6C97CF6728C1. The online version of this work is archived and available from the following digital repositories: PeerJ, PubMed Central and CLOCKSS.

## RESULTS

### Systematics

Order Tanaidacea Dana, 1849
Suborder Apseudomorpha Sieg, 1980
Superfamily Apseudoidea Leach, 1814
Family Metapseudidae Lang, 1970
Subfamily Chondropodinae Guţu, 2008

**Diagnosis (after *Guţu, 2008*).** Body dorsoventrally flattened, with small lateral or anterolateral acute or rounded processes at the level of pereonites. Pleon with five short pleonites (with or without lateral plumose setae) and a short pleotelson (with a small acute lateral process in the first half). Antennule peduncle usually with at least one conspicuous denticle on the inner margin of the first article; at least outer flagellum long, multi-segmented. Antenna peduncle with second article long, in rare cases without spiniform denticles on the inner margin; squama present (small). Mandible with tri-articled palp. Maxillule with bi-articled palp. Maxilliped with third article of the palp evidently longer than broad, sometimes shorter than the second one; second article of the palp with a great outerodistal spine and exceptionally with numerous plumose setae on the outer side. Cheliped obvious dimorphic or not, with exopod. Pereopod-1 well-developed, much larger than the following pereopods, with exopod; basis thick with long plumose setae which alternate with spiniform denticles on the dorsal margin (in rare cases, denticles absent); sometimes coxa with a spiniform prolongation (more or less developed). Pereopods 2–6 thin, different from pereopod-1; pereopods 2, 3 and 5 with long and similar propodus (much longer than the carpus, the last of about the same length with the merus), slightly curved, with at most four sternal spines; dactylus also thin and long, with a well-developed claw. Pleopods present, biramous, in three or five pairs. Uropod with multi-segmented (but not long) rami.

**Genera included:** *Bamberus* Stępień & Błażewicz-Paszkowycz, 2013; *Calozodion* Gardiner, 1973; *Chondropodus* Guţu, 2006; *Hoplopolemius* Sganga & Roccatagliata, 2016; *Julmarichardia* Guţu, 1989; *Trichapseudes* Barnard, 1920; *Vestigiramus* Guţu, 2009; *Zaraza* Guţu, 2006.

**Remarks**

Chondropodinae is considered a monophyletic taxon with precise and consistent diagnosis among the included genera (*Guțu, 2008*, *2009*; *Stępień & Błażewicz-Paszkowycz, 2013*); it has been suggested as a separate family (*Stępień & Błażewicz-Paszkowycz, 2013*). The definition of Chondropodinae by *Guțu (2008*, *2009)*, supplemented by *Stępień & Błażewicz-Paszkowycz (2013)*, is based on the setation of the pereopod-1 basis (row of plumose setae on dorsal margin), the relative carpus-to-merus length of pereopods 2, 3 and 5, and the length of the propodus, which appears clearly longer than the carpus and has at least four sternal spines.

This coherent definition of the subfamily was disrupted after the classification of the genus *Deidamiapseudes* Sganga & Roccatagliata, 2016 within Chondropodinae (classification provided by *WoRMS, 2019*). In contrast to the other Chondropodinae, the genus *Deidamiapseudes* lacks plumose setae on the pereopod-1 basis, whereas the propodus of pereopods 2, 3 and 5 is shorter than the carpus; pereopods 2–3 propodus has seven to eight ventral spines.

Intriguingly, *Sganga & Roccatagliata (2016)* decided to neither classify their genus to any subfamily nor even to the apseudoidean family and noted that *Deidamiapseudes* shows some morphological similarities to both Apseudidae and Metapseudidae, such as an apophysis on the coxa of pereopod-1, recorded in both families, large spines on the inner margin of the antennule peduncle article-1 found in Metapseudidae, but pereopods "adapted for walking", which is characteristic for Apseudidae but not for Metapesudidae. Because *Deidamiapseudes* is a morphologically clearly distinct from other Chondropodinae, we removed this genus from the subfamily and placed in Apseudoidea, with an uncertain family status (*incertae sedis*).

Genus *Muvi* gen. nov.
urn:lsid:zoobank.org:act:60F20E13-CC0C-4779-828F-50A561E1BB85

**Diagnosis.** Rostrum triangular pointed, lateral margin smooth. Eyelobes with visual elements. Pereonites wider than long. Antennule peduncle of four articles, article-1 with only a single apophysis on the outerodistal corner; flagella equal in length, each with twelve segments. Maxillule inner lobe well-developed. Labial palp with three distal setae. Exopod on cheliped and pereopod-1 well-developed, with nine and eleven plumose setae, respectively. Pereopod-1 coxa with distinct apophysis. Bases of pereopods 1–6 without apophyses. Pereopod-1 propodus 1.5 times as long as wide. Pleopods in five pairs. Uropod basal article without hyposphaenium; endopod of seven segments, exopod of three segments.

**Type species:** *Muvi schmallenbergi* gen. nov., sp. nov. (by monotypy).

**Etymology.** The name is an acronym for the Museum of Victoria (Melbourne, Australia), where the type species of the new genus is deposited.

**Remarks**

*Muvi* gen. nov. is classified to the subfamily Chondropodinae, based on a combination of the following subfamily characters: pleon of five free pleonites, antennule peduncle article-1 with apophysis, antenna peduncle article-2 elongated, mandibular palp of three articles and pereopod-1 basis with a row of plumose setae dorsally. A multi-segmented inner flagellum of antennule with the same size and segment number as the outer flagellum is the main character distinguishing the new genus from other Chondropodinae. Moreover, *Muvi* differs from the following genera:

- *Bamberus* by pleotelson without lateral process, antennule article-1 with single apophysis, well-developed inner lobe of maxillule, labial palp with three setae distally and uropod basal article without apophysis. In *Bamberus*, the pleotelson has a single process on the lateral sides, antennule article-1 lacks apophyses, the inner lobe of maxillule is reduced and bears only two distal setae, the labial palp has two setae, and the uropod basis has a distinct apophysis (*Stępień & Błażewicz-Paszkowycz, 2013*);
- *Calozodion* by having three strong setae distally on labial palp; there is only one distal spine in *Calozodion* (*Guţu, 2002*);
- *Chondropodus* by having eyelobes with visual elements, pereonites with lateral processes, a pleotelson without a lateral process, a labium with three distal setae, pereopod-1 coxa with distinct apophysis, propodus only 1.5 L:W, pleopod exopod with a single article. In *Chondropodus*, the eyelobes lack visual elements, pereonites do not have lateral processes, labial palp has a single distal spine, pereopod-1 lacks coxal apophysis, and propodus is at least twice as long as wide; the pleopod exopod is bi-articled (*Guţu, 2006a*);
- *Hoplopolemius* by having antennule peduncle article-1 with only one apophysis. Article-1 of the antennule peduncle in *Hoplopolemius* has clearly more than one apophysis (*Richardson, 1902*; *Guţu, 2002*; *Larsen & Shimomura, 2006*);
- *Julmarichardia* by having a rostrum with smooth lateral edges and antennule peduncle article-1 with a single apophysis. The lateral edges of the *Julmarichardia* rostrum are distinctly serrated, and the antennule peduncle article-1 in members of this genus has more than one apophysis (*Barnard, 1914*; *Guţu, 1989a*, *1989b*; *Bamber & Sheader, 2005*; *Ritger & Heard, 2007*);
- *Trichapseudes* by having five pairs of pleopods; in *Trichapseudes*, only three pairs of pleopods are present (*Barnard, 1920*);
- *Vestigiramus* by having well-developed and setose cheliped exopod. *Vestigiramus* has a reduced uni-articled and naked cheliped exopod (*Guţu, 2009*);
- *Zaraza* by three setae distally on the labial palp and five pairs of pleopods. *Zaraza* has a labial palp with a single terminal seta (spine) and three pairs of pleopods (*Guţu, 2006b*).

## Key to the genera of the subfamily Chondropodinae (modified after *Guţu, 2008*)

**1** - Rostrum with marginal denticles . . . . . . . . . . . . . . . . . . *Julmarichardia* Guţu, 1989

- Rostrum without marginal denticles . . . . . . . . . . . . . . . . . . . . . . . . . . . . . 2

**2** - Pereopod-1 propodus cylindrical, much longer than thick or the length of the carpus. . . . . . . . . . . . . . . . . . . . . . . . . . . . . . . . . . . . . . . . . . . . . . . 3

- Pereopod-1 propodus wide, not much longer than thick or the length of the carpus . . . . . . . . . . . . . . . . . . . . . . . . . . . . . . . . . . . . . . . . . . . . . . 4

**3** - Uropod peduncle with strong apophysis dorsally . . . . . . . . . . . . . . . . . . . . . . . . . . . . . . . . . . . . . *Bamberus* Stępień & Błażewicz, 2013

Uropod peduncle without strong apophysis dorsally . . . . . *Chondropodus* Guţu, 2006

**4** - Antennule inner flagellum bi-segmented . . . . . . . . . . . . . . . . . . . . . . . . . . . . 5

- Antennule inner flagellum multi-segmented . . . . . . . . . . . . . . . . . . . . . . . . . . 6

**5** - Cheliped exopod tri-articled, with terminal setae . . . . . . *Calozodion* Gardiner, 1973

Cheliped exopod uni-articled, without terminal setae . . . . . *Vestigiramus* Guţu, 2009

**6** - Antennule inner flagellum equal in length to outer flagellum . . . . . . *Muvi* gen. nov.

Antennule inner flagellum shorter than outer flagellum . . . . . . . . . . . . . . . . . . 7

**7** - Pereopod-1 exopod with last article round (and large), having more than 20 plumose marginal setae . . . . . . . . . . . . . . . . . . . . . . . . . . . . . *Trichapseudes* Barnard, 1920

- Pereopod-1 exopod with last article normal (elongated), having clearly less than 20 plumose marginal setae . . . . . . . . . . . . . . . . . . . . . . . . . . . . . . . . . . . . . 8

**9** - Five pairs of pleopods . . . . . . . . . . . . *Hoplopolemius* Sganga & Roccatagliata, 2016

- Three pairs of pleopods . . . . . . . . . . . . . . . . . . . . . . . . . . . . . . *Zaraza* Guţu, 2006

*Muvi schmallenbergi* sp. nov.
urn:lsid:zoobank.org:act:14743564-C6F2-42CE-A181-CE30F5C5A2C2
(Figs. 1–3)

**Material examined.** Holotype female (MNV J74649), 4.5 mm, SLOPE 99, Victoria, Off Portland, 38° 31′ 34″ S, 141° 58′ 46″ E, depth 49.6 m, 11 May 1994, Smith-McIntyre grab, coll. G.C.B. Poore. Paratype, female (MNV J74648), 5.9 mm, the same locality, dissected on slides.

**Diagnosis.** As for the genus.

**Etymology.** The new species is dedicated to Barbara Schmallenberg.

**Description of female (body of holotype, appendages from paratype)**
Body (Figs. 1A, 1B) 4.5 mm long. Cephalothorax 21% of total body length; rostrum triangular and pointed (Figs. 1A, 2C); eyelobes pointed with visual elements (Figs. 1A, 1B). Pereon 47% of total body length; pereonites length-width ratio 0.3, 0.4, 0.5, 0.5, 0.5 and 0.3; pereonites 2–5 with dorsoproximal apophyses on lateral margin. Pleon 32% of total body length; pleonites equal in length, about 0.2 L:W, with pointed lateral margins; pleotelson just longer than the last three pleonites combined.

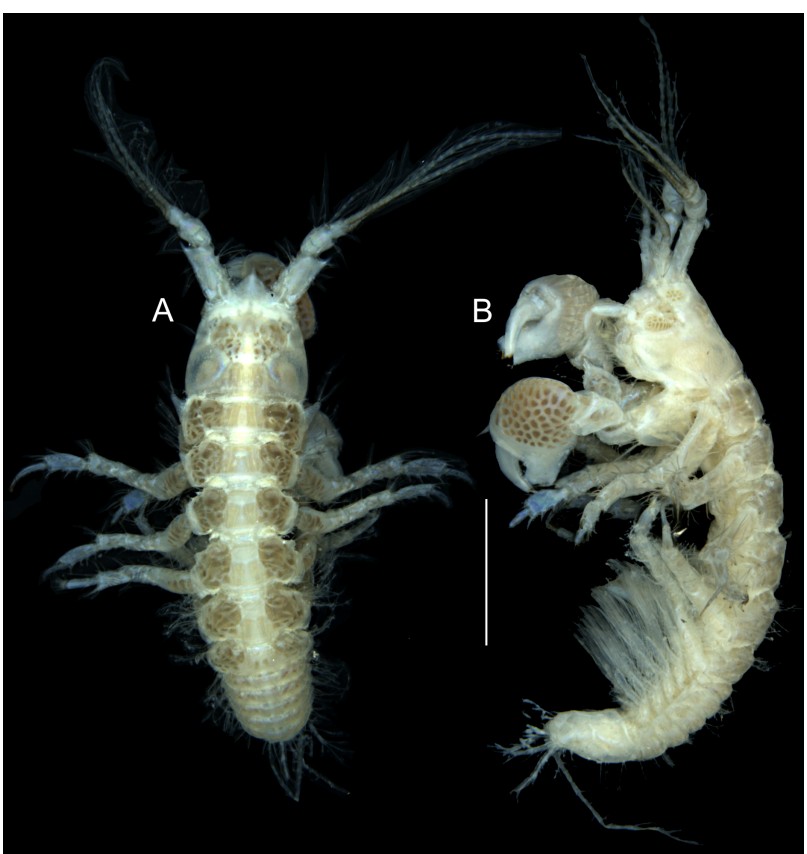

**Figure 1** *Muvi schmallenbergi* **sp. nov. holotype female (cat. no. MNV J74649), length 4.5 mm.**
Habitus illustration. (A) Body dorsal view. (B) Body lateral view. Scale bar = 1 mm. Photographs:
Magdalena Błażewicz.                                                                               

Antennule (Fig. 2A) peduncle article-1 2.1 L:W and 1.9x article-2, with four simple and
two penicillate setae on the inner margin and one simple, one plumose seta, and four
penicillate setae on the outer margin; single apophysis present in the outerodistal corner;
article-2 1.2 L:W and 1.9x article-3, with five simple and two penicillate setae subdistally;
article-3 as long as wide, with two simple setae distally; common article short and
naked; flagella subequal, each with 12 segments, setation as figured (no aesthetasc present).

Antenna (Fig. 2B) peduncle article-1 short and naked; peduncle article-2 1.7 L:W and
3x article-3, with three minute distal and subdistal setae; squama narrow, 4.5 L:W, with
simple subdistal seta and three distal setae; peduncle article-3 0.8 L:W and 0.7x article-4,
with two long simple setae distally; article-4 1.2 L:W and 0.9x article-5, with three
penicillate distal setae; article-5 1.4 L:W, with two short simple, two long simple, and two
penicillate setae distally, and one mid-length simple seta; flagellum of six segments,
setation as figured.

Mouthparts. Labrum (Fig. 2D) rectangular with numerous minute setae on distal and
lateral margins. Right mandible (Fig. 2E) incisor with four well-calcified triangular
teeth; outer margin with tubercles and with minute setae distally. Left mandible (Fig. 2F)
outer margin with distally setulose tubercles; incisor with four distal teeth; *lacinia mobilis*

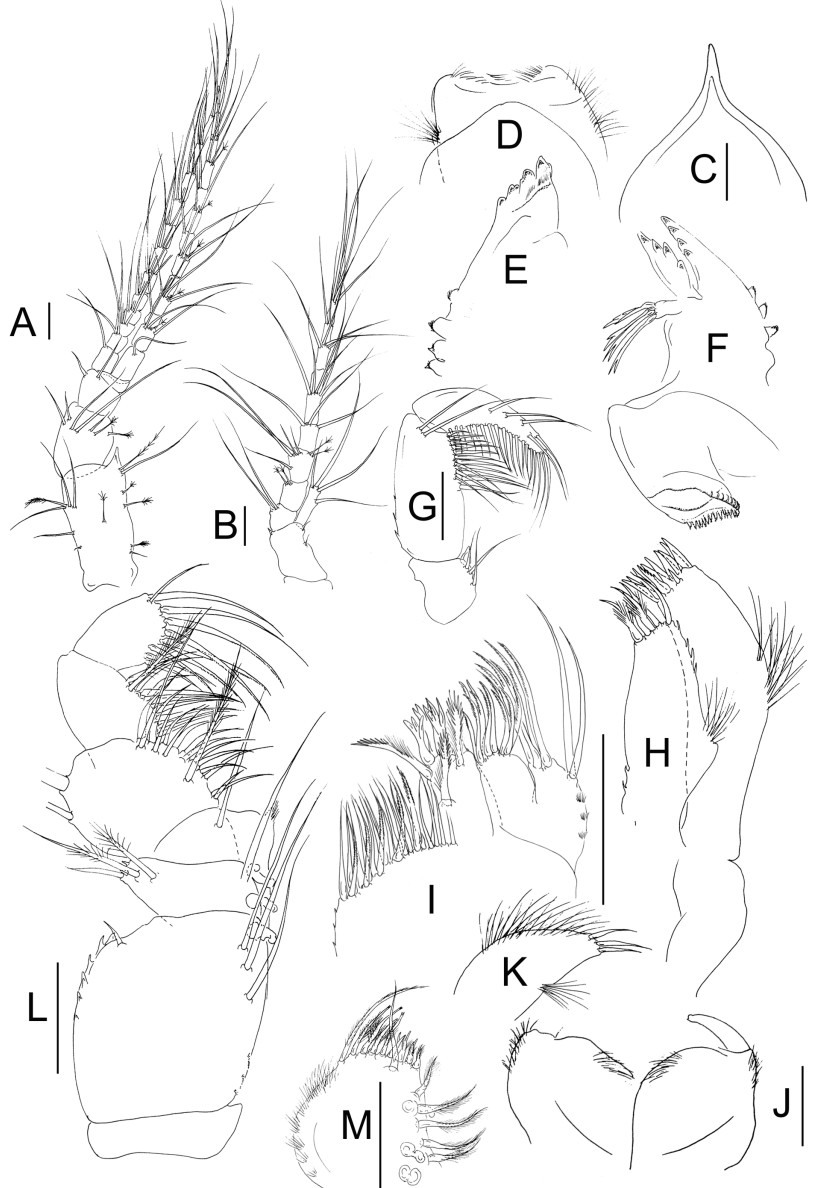

**Figure 2 *Muvi schmallenbergi* sp. nov. paratype female (cat. no. MNV J74648).** Antennule, antenna, and mouth parts illustrations. (A) Antennule. (B) Antenna. (C) Rostrum. (D) Labrum. (E) Right mandible. (F) Left mandible. (G) Mandibular palp. (H) Maxillule. (I) Maxilla. (J) Labium. (K) Labial palp. (L) Maxilliped. (M) Maxillipedal endite. Scale bars = 0.1 mm.

as long as incisor, with four teeth, setiferous lobe with four complex-tip setae; molar broad, distally serrated; palp (Fig. 2G) article-1 1.1 L:W, with five inner setae; article-2 2.1 L:W with outer margin serrated, two simple distal setae and row of about 14 inner setae starting from the middle of the article, decreasing in length towards the distal end of the article; article-3 3.7 L:W, with five outer setae and a row of about 18 setae along the inner margin. Maxillule (Fig. 2H) inner endite with five setae distally (at least three setulated), inner and outer margins serrated, outer margin with tuft of setae and

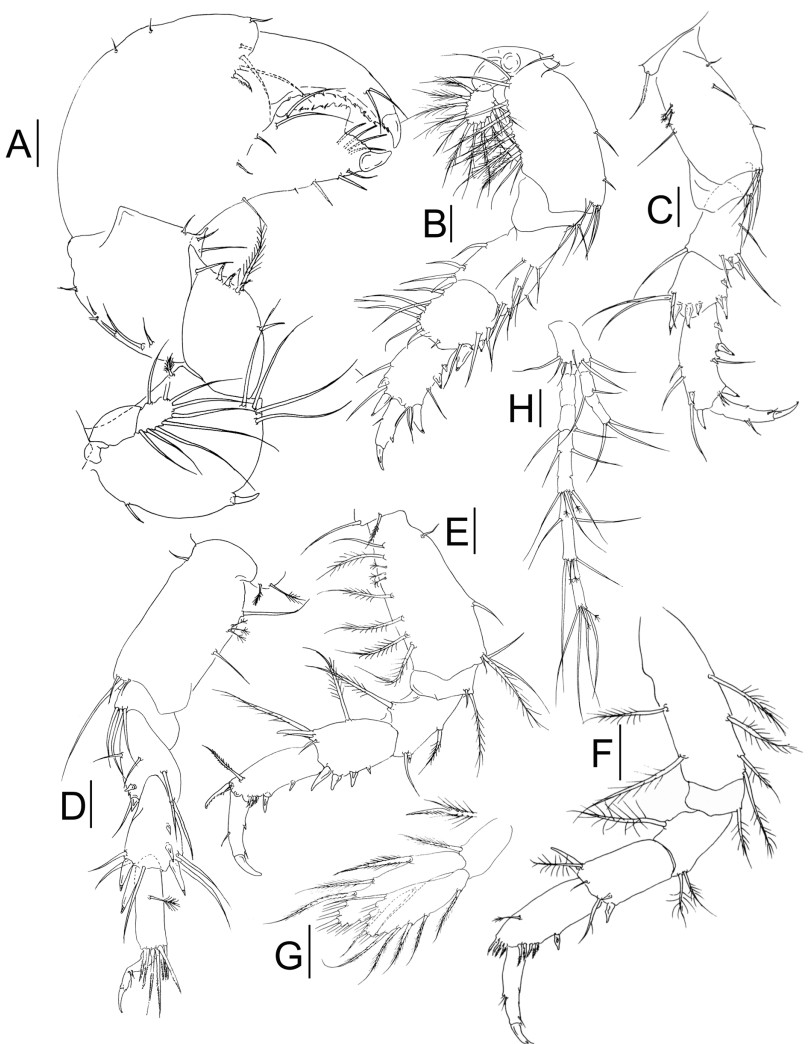

**Figure 3** *Muvi schmallenbergi* **sp. nov. paratype female (cat. no. MNV J74648).** Cheliped and pereopods illustrations. (A) Cheliped. (B) Pereopod-1. (C) Pereopod-3. (D) Pereopod-4. (E) Pereopod-5. (F) Pereopod-6. (G) Pleopod. (H) Uropod. Scale bars = 0.1 mm.

subproximal tubercle; outer endite with eleven spines (one appearing serrated), outer margin setulated. Maxilla (Fig. 2I) outer lobe of moveable endite with two subdistal simple setae and six distal serrated setae, outer margin with microtrichiae; inner lobe of moveable endite with ten serrated setae distally; outer lobe of fixed endite distally with three tri-furcated and three serrated setae; inner lobe of fixed endite with 22 setae (at least five of them serrated). Labium (Figs. 2J, 2K) outer and inner margins with setation distally; palp (Fig. 2K) lateral margins setulated, distally with three simple spines.

Maxilliped (Fig. 2L) coxa naked. Basis 1.1 L:W, with laterodistal seta and three subdistal long setae, additionally outer margin with teeth and inner margin with proximal microtrichiae. Palp article-1 0.4 L:W, with long simple seta on inner margin and one simple seta and two plumose setae on outer margin; article-2 1.2 L:W, with two rows of setae on inner margin (first with 11 setae and second with five setae, at least one seta

plumose), and two long laterodistal setae; article-3 about as long as wide, with row of about 10 setae along inner margin (at least one plumose); article-4 1.3 L:W, with row of nine distal setae (at least five of them serrated) and one outer seta. Endite (Fig. 2M) outer margin setulated; inner margin with three coupling hooks and five short plumose setae; distal margin with nine short plumose setae/spines (some with complex tip) and one long plumose seta; subdistal seta simple (not leaf-like).

Cheliped (Fig. 3A) basis 1.1 L:W, with plumose dorsodistal seta, simple seta ventroproximally, one spine ventrally at mid-length and four long setae ventrodistally; exopod of three articles, article-3 with nine plumose setae (drawn as simple for figure clarity); merus 1.2 L:W and 0.7x basis, with one simple seta, one plumose seta, four spines and apophysis ventrally; carpus 0.9 L:W and 1.2x merus, with row of seven setae along dorsal margin, three setae ventrally and distal apophysis; propodus 0.9 L:W and 1.8x carpus, with three short setae dorsally, three setae on outer surface, and one serrated inner spine near dactylus insertion; fixed finger about 0.8x propodus, with two proximal outer setae, four setae ventrally, cutting margin with two long setae near dactylus insertion, four setae distally and small teeth accompanied with minute setae in proximal half; dactylus just longer than fixed finger with two subdistal setae, and row of teeth and spinules along the cutting edge.

Pereopod-1 (Fig. 3B) coxa long and with smooth margins, with two setae; basis 2.2 L:W and 2.0x merus, with two ventral setae and four ventrodistal setae, row of plumose (five) and simple (five) setae along dorsal margin; exopod of three articles, article-3 with 11 plumose setae; ischium with three ventrodistal setae; merus 1.3 L:W and 1.1x carpus, with one dorsodistal spine, one ventrodistal spine, four ventral and three dorsal setae, and four setae on inner surface; carpus 1.1 L:W and 1.1x propodus, with two ventrodistal spines, three setae ventrally, one short seta distally and six setae dorsally; propodus 1.5 L:W and 1.5x dactylus, with three spines and four setae ventrally, and with two spines and two setae dorsally; dactylus 2.9 L:W and x 2.0 unguis, with one dorsal seta and ventral apophysis; dactylus and unguis combined as long as propodus.

Pereopod-2 as pereopod-3 (not figured).

Pereopod-3 (Fig. 3C) coxa with one seta; basis 2.1 L:W and 1.8x merus, with tuft of three setae ventrodistally, one simple seta, three penicillate setae dorsally, and three simple setae ventrally; ischium with two ventrodistal setae; merus 1.7 L:W and 1.3x carpus, with one small spine and one larger spine ventrodistally, one dorsodistal seta, one ventral seta, and two ventrodistal setae; carpus 1.2 L:W and 0.65x propodus, with one ventral subproximal spine, two ventrodistal spines, one short seta, two long setae, and three spines dorsodistally; propodus 2.5 L:W and 1.5x dactylus, with three ventroproximal spines, one ventrodistal spine, one dorsal subdistal spine, one dorsodistal spine, one ventral seta, two short ventrodistal setae, two dorsal setae, and one dorsodistal seta; dactylus 3.2 L:W, with ventral tooth, ventrodistal seta, and two dorsal setae; unguis about 0.5x dactylus; together about as long as propodus.

Pereopod-4 (Fig. 3D) coxa with two penicillate setae; basis 2.1 L:W and 2.0x merus, with one ventroproximal seta, two short setae, and one long seta ventrodistally, and two simple setae and three penicillate setae dorsally; ischium with three ventrodistal

setae; merus 1.5 L:W and 0.9x carpus, with two short ventrodistal spines, two ventral setae, one ventrodistal seta, and two long dorsodistal setae; carpus 1.8 L:W and as long as propodus, with two short spines and one longer dorsal spine, two ventrodistal spines, two ventrodistal setae, and three dorsodistal setae; propodus 3.3 L:W and 2.0x dactylus, with one penicillate dorsal seta and eight serrated setae distally; dactylus damaged, 2.0x unguis, with two short setae ventrally.

Pereopod-5 (Fig. 3E) coxa with one simple seta and one plumose seta; basis 2.3 L:W and 2.2x merus, with simple ventroproximal seta, one mid-length ventral seta, one plumose seta and three simple setae ventrodistally, dorsal margin with one simple seta, five plumose setae and three penicillate setae; ischium with one plumose seta and one short simple seta ventrodistally; merus 1.7 L:W and as long as carpus, with one short ventrodistal spine, one plumose ventral seta, one simple ventrodistal seta, and one plumose dorsodistal seta; carpus 1.5 L:W and 0.8x propodus, with four spines increasing in size along ventral margin, and one dorsodistal spine, one plumose dorsal seta, and one plumose seta and two simple dorsodistal setae; propodus 2.0 L:W and 1.7x dactylus, with two ventral spines, three ventrodistal serrated minute spines, one serrated dorsodistal spine, and one dorsodistal plumose seta; dactylus with mid-length minute ventral spine, medially and dorsally with one seta, combined length of dactylus with that of unguis 0.9x propodus.

Pereopod-6 (Fig. 3F) basis 3.1 L:W and 3.4x merus, with two dorsal plumose setae and four ventral plumose setae; ischium with one ventrodistal plumose seta; merus 1.3 L:W and 0.6x carpus, with one subdistal plumose seta dorsally and two ventrodistal plumose setae; carpus 1.9 L:W and 0.9x propodus, with two ventrodistal spines, one dorsodistal spine, one ventrodistal simple seta, one distal simple seta, and one dorsodistal plumose seta; propodus 2.2 L:W and 1.2x dactylus, with one mid-length spine ventrally, and at least nine minute serrated spines ventrodistally and dorsodistally, and one dorsal mid-length penicillate seta; dactylus with ventrodistal seta, ventral serration, and three dorsal setae; dactylus with mid-length minute ventral spine, medially and dorsally with one seta, combined length of dactylus with that of unguis 1.1x propodus.

Pleopods (Fig. 3G) basal article 1.7 L:W, with one distal plumose seta; endopod just longer than exopod, with two plumose setae subdistally, seven plumose setae distally, and one dorsal seta, and one ventral mid-length seta; exopod with eleven plumose setae along distal end and one plumose ventroproximal seta.

Uropod (Fig. 3H) basal article 1.8 L:W, with seven simple setae distally; exopod of three segments, segment-2 with two distal setae, segment-3 with three distal setae; endopod 3.7x exopod, of seven segments, some with mid-length setae apparently indicating segment fusion; segment-3 with two distal setae; segment-4 with four mid-length setae, and two penicillate setae, and three simple setae distally; segment-5 with two mid-length setae, and two penicillate setae, and two simple setae distally; segment-7 with two penicillate setae and four simple setae distally; other segments naked.

**Distribution.** *Muvi schmallenbergi* is known only from the type locality, offshore Portland (Australia), from a depth of 49.6 m and habitats rich in red algae.

**Table 1 Depth and sediment type for known Chondropodinae species.**

| Species | Depth (m) | Sediment | References |
|---|---|---|---|
| *Bamberus jinigudirus* Stępień & Błażewicz-Paszkowycz, 2013 | 4–12 | Sand, fine rubble in groove, dead Acropora, finger rubble | *Stępień & Błażewicz-Paszkowycz, 2013* |
| *Calozodion bacescui* Guţu, 1996 | 29–50 | Sandy substratum with biogenic gravel, limestone concretions and algae | *Guţu, 1996* |
| *Calozodion bogoescui* Guţu, 2014 | Shallow waters | – | *Guţu, 2014* |
| *Calozodion dominiki* Bochert, 2012 | 26–117 | – | *Bochert, 2012* |
| *Calozodion heardi* Guţu, 2002 | – | – | *Guţu, 2002* |
| *Calozodion moyas* Menioui, 2013 | 6 | – | *Menioui, 2013* |
| *Calozodion multispinosum* Guţu, 1984 | 22 | Dark grey mud | *Guţu, 1984* |
| *Calozodion pabisi* Jakiel & Jóźwiak, 2015 | 386 | *Lophelia* reef | *Jakiel et al., 2015* |
| *Calozodion simile* Guţu, 2006 | – | – | *Guţu, 2006a* |
| *Calozodion singularis* Guţu, 2002 | – | – | *Guţu, 2002* |
| *Calozodion suluk* Bamber & Sheader, 2005 | 23–35 | 2% gravel, 75–78% sand, 9–12% silt, 10–11% clay | *Bamber & Sheader, 2005* |
| *Calozodion tanzaniense* Guţu, 2014 | Shallow waters | – | *Guţu, 2014* |
| *Calozodion wadei* Gardiner, 1973 | 6.1 | Fine sand, silt and clay | *Gardiner, 1973* |
| *Chondropodus curvispinus* Guţu, 2006 | – | – | *Guţu, 2006a* |
| *Chondropodus rectispinus* Guţu, 2006 | – | – | *Guţu, 2006a* |
| *Hoplopolemius propinquus* (Richardson, 1902) | – | – | *Richardson, 1902* |
| *Hoplopolemius toyoshious* (Larsen & Shimomura, 2006) | 73 | Shell sand | *Larsen & Shimomura, 2006* |
| *Hoplopolemius triangulatus* (Richardson, 1902) | – | – | *Richardson, 1902* |
| *Julmarichardia alinati* Guţu, 1989 | 6–450 | – | *Guţu, 1989b* |
| *Julmarichardia bajau* Bamber & Sheader, 2005 | 23–35 | 2% gravel, 75–78% sand, 9–12% silt, 10–11% clay | *Bamber & Sheader, 2005* |
| *Julmarichardia deltoides* (Barnard, 1914) | 90 | – | *Barnard, 1914* |
| *Julmarichardia dollfusi* (Guţu, 1989) | – | – | *Guţu, 1989a* |
| *Julmarichardia gutui* Ritger & Heard, 2007 | 78–83 | – | *Ritger & Heard, 2007* |
| *Julmarichardia thomassini* Guţu, 1989 | 250 | – | *Guţu, 1989b* |
| *Muvi schmallenbergi* sp. nov. | 49.6 | – | |
| *Trichapseudes tridens* Barnard, 1920 | 31–155 | – | *Barnard, 1920* |
| *Vestigiramus antillensis* Guţu, 2009 | 1–2 | Dead corals and seagrass beds | *Guţu, 2009* |
| *Vestigiramus codreanui* (Guţu, 1996) | 29 | Limestone concretions and algae | *Guţu, 1996* |
| *Vestigiramus* sp. Araújo-Silva & Larsen, 2012 | 71.6 | Sandy bottom associated with sponge and algae | *Araújo-Silva & Larsen, 2012* |
| *Zaraza linda* Guţu, 2006 | 0.5–2 | Dead corals covered with algae | *Guţu, 2006b* |

## DISCUSSION

### Distribution of Chondropodinae

#### Depth and habitat

Chondropodinae, as the other metapseudids, are mostly represented by shallow-water taxa whose vertical distributions rarely exceed the edge of the continental shelf (Table 1). To date, only three species have been recorded from greater depths, namely *Julmarichardia*

*thomassini* Guţu, 1989 found at 250 m, *Calozodion pabisi* Jakiel & Jóźwiak, 2015 found at 386 m, and *Julmarichardia alinati* Guţu, 1989 recovered in a wide depth range from 6 to 450 m (Table 1). Chondropodinae and other members of the family Metapseudidae are usually associated with coral reefs or hydroid colonies (*Sieg, 1986a*; *Stępień & Błażewicz-Paszkowycz, 2013*), but also occur in or on a variety of different substrata, *e.g.*, sand, silt clay, rubble, algae, or dead corals (Table 1).

### Geographical distribution

Chondropodinae are most diversified in the Atlantic and represented by 17 species. The highest number of species (six) has been recorded for the relatively restricted areas of the Eastern Gulf of Mexico, Cuba, Haiti, and Southeast Mexico (*Richardson, 1902*; *Gardiner, 1973*; *Guţu, 1984*, *2002*, *2006b*, *2009*). Furthermore, five Chondropodinae are known from East African coasts (*Barnard, 1914*, *1920*; *Guţu, 1989b*, *2014*), and four species were found in West Africa (*Guţu, 2006a*; *Menioui, 2013*). The genera of Chondropodinae are widely distributed and usually present in more than one marine basin. For example, *Julmarichardia* was found in the Mozambique Channel, on the Malaysian coast, in Northwest Australia, and in the Northeast Atlantic. *Calozodion* was found on the Brazilian coast, off Angola, off Mauritania, in Malaysia, in the Caribbean Sea, and in the Gulf of Mexico (Figs. 4 and 5). Two non-monotypic genera of Chondropodinae, *Vestigiramus* and *Chondropodus*, have a restricted distribution. *Vestigiramus* is represented by three species that occur along the East coast of South America, and *Chondropodus*, with two species, has been recorded from the coast of Mauritania. The distribution of Chondropodinae is clearly limited to tropical, subtropical, and, to some extent, temperate waters; likewise, they are entirely absent in higher latitudes (Figs. 4 and 5). The most northerly record is held by *Julmarichardia dollfusi* situated in the subboreal zone (latitude 49° N) off Jersey in the Northeast Atlantic (English Channel) (*Guţu, 1989a*). Until now, the most southerly records of Chondropodinae have been made by Barnard, who described *Julmarichardia deltoides* from the Great Fish Point Lighthouse (Port Alfred, South Africa) and *Trichapseudes tridens* offshore of East London (South Africa) (approximately 33°S; coordinates are not specified in the original descriptions) (*Barnard, 1914*; *Barnard, 1920*). As of this record, *Muvi schmallenbergi* is the southernmost record of the subfamily, found at approximately 38°S. The Atlantic sectors of the Arctic and the Antarctic are among the most studied areas of the World Ocean (*Sieg, 1986b*; *1986c*; *Bird, 2010*; *Błażewicz-Paszkowycz, 2014*; *Jakiel, Stępień & Błażewicz, 2018*); thus, the absence of the Chondropodinae in polar regions cannot be explained by biased sampling efforts.

The longitudinal gradient in diversity, observed in many groups of marine invertebrates (decapods, gastropods, and bivalves) or vertebrates (*e.g.*, fishes), point to a peak in the tropics and a decrease towards the poles (*Clarke & Crame, 2010*; *Rabosky et al., 2018*). The highest diversity around the equator is justified by the larger area and the longer evolutionary time of the tropical regions, where diversity is supported by higher rates of

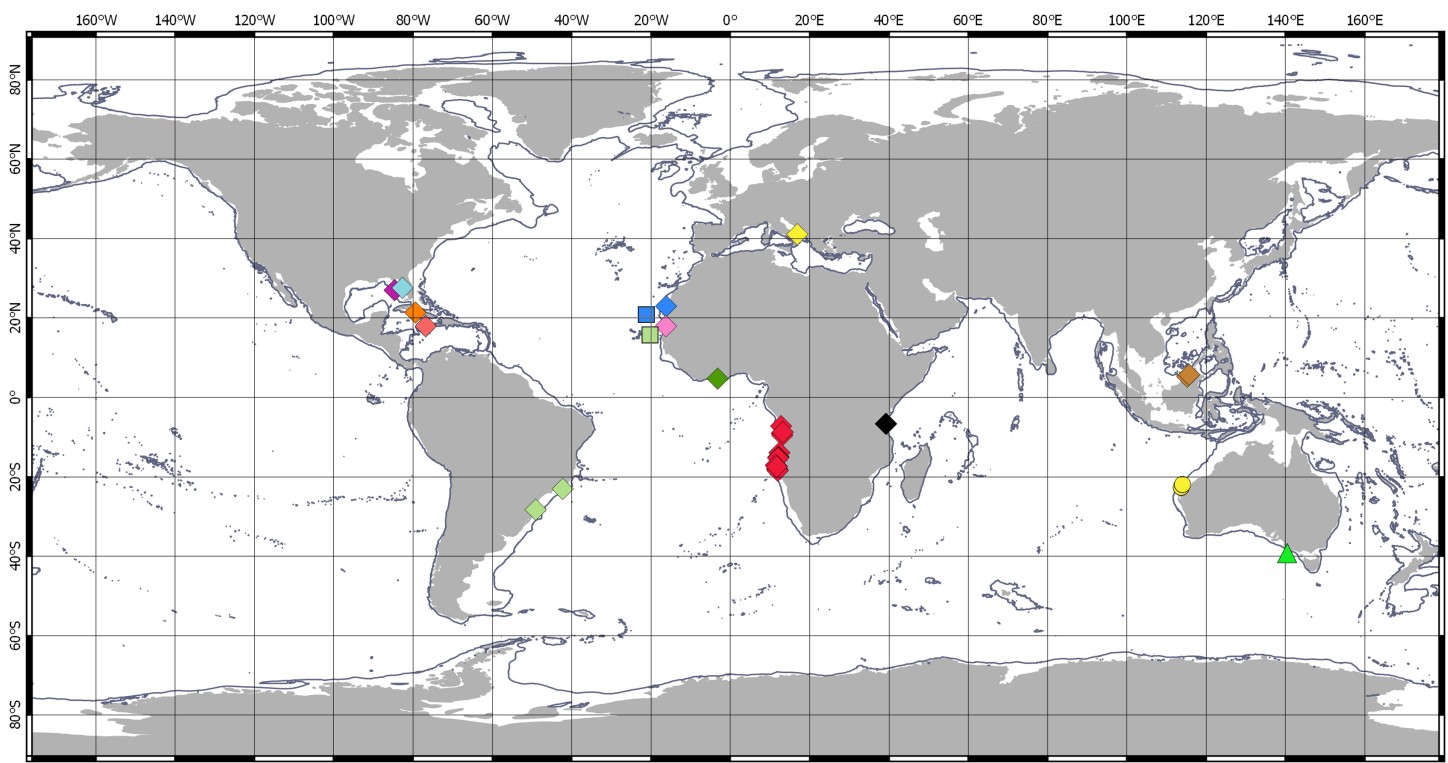

**Figure 4 Distribution of Chondropodinae (1). Circle—genus *Bamberus* represented only by *B. jinigudirus*. Triangle—genus *Muvi* represented only by *M. schmallenbergi*. Diamond—genus *Calozodion*:** light green—*C. bacescui*; yellow—*C. bogoescui*; red—*C. dominiki*; purple—*C. heardi*; blue —*C. moyas*; orange—*C. multispinosum*; green—*C. pabisi*; pink—*C. simile*; light blue—*C. singularis*; brown—*C. suluk*; black—*C. tanzaniense*; light red—*C. wadei*. **Square—genus *Chondropodus*:** blue—*Ch. curvispinus*; green—*Ch. rectispinus*.

speciation and lower extinction rates (*Mittelbach et al., 2007*; *Brown, 2014*). Cenozoic glaciations interrupted by interglacial events are stated as the main factor responsible for extinctions of shallow-water taxa in polar regions (*Clarke & Crame, 2010*; *Thatje, 2012*). Recolonization of the vacant habitats of the Antarctic shelf is assumed to be limited for tropic or temperate fauna because of their physiological adaptation to warmer waters (*Thatje, 2012*; *Brown, 2014*). Hence, *Brown (2014)* remarked that tropical species and lineages have a long evolutionary history in relatively equable environments and may not tolerate several abiotic stresses (*e.g.*, low temperature and extreme seasonality).

The pantropical and pantemperate distribution of Tanaidacea and their absence in high latitudes has previously been reported for most shallow-water families of Apseudomorpha (*Błażewicz-Paszkowycz, 2014*) and some plesiomorphic families of Tanaidomorpha, such as Tanaididae, Pseudozeuxidae, Paratanaidae, and Leptocheliidae (*Błażewicz-Paszkowycz, Bamber & Anderson, 2012*). *Sieg (1992)* stated that tropical or temperate Tanaidacea that became extinct during glaciations were replaced by deep-sea taxa because of similar temperature regimes in the abyssal and the Antarctic shelf (polar emergence). He argued that tanaidaceans which evolved before the Eocene were theoretically "ready" to colonize vacant Antarctic habitats and that the Antarctic shallow-water tanaidaceans are dominated by phylogenetically young taxa which lack functional eyes, evidencing their deep-sea origin (*Sieg, 1992*). *Błażewicz-Paszkowycz (2014)*

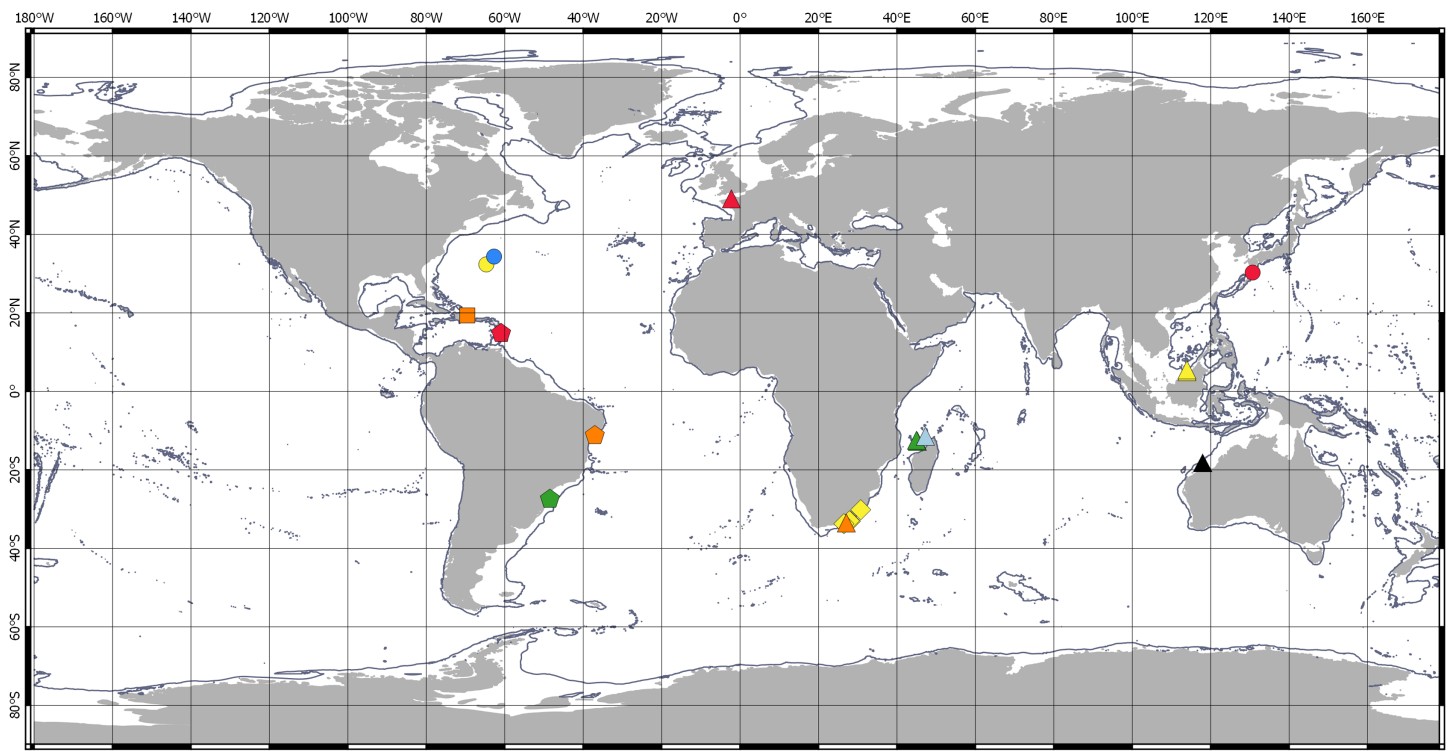

**Figure 5  Distribution of Chondropodinae (2). Circle—genus *Hoplopolemius*:** yellow—*H. propinquus*; red—*H. toyoshious*; blue—*H. triangulatus*. **Triangle—genus *Julmarichardia*:** green—*J. alinati*; yellow—*J. bajau*; orange—*J. deltoides*; red—*J. dollfusi*; black—*J. gutui*; blue—*J. thomassini*. **Diamond—genus *Trichapseudes*** represented only by *T. tridens*. **Pentagon—genus *Vestigiramus*:** red—*V. antillensis*; green—*V. codreanui*; orange —*Vestigiramus* sp. *Araújo-Silva & Larsen, 2012*. **Square—genus *Zaraza*** represented only by *Z. linda*.

has supplemented Sieg's hypothesis by indicating that some tanaidaceans might have survived the glaciations in the deeper shelf or slope refugia or colonized the Antarctic *via* the Scotia Arc. This idea is strongly supported by the presence of representatives of typically tropical families in Antarctica, such as *Paratanais oculatus* (Paratanaidae), *Allotanais hirsutus* (Tanaididae), and *Synapseudes aflagellatus* (Metapseudidae).

The Arctic is less thermally isolated than the Antarctic, which is surrounded by the Antarctic Circumpolar Current (ACC). Furthermore, it is supported by the transitional zones to the south, where the temperate Atlantic waters merge and mix with the polar waters (*Loeng et al., 2005*; *Stepanjants et al., 2006*). Also, warm water masses transported to the Arctic by the Atlantic currents promote the distribution of temperate fauna to the polar zone (*McBride et al., 2014*; *Csapó, Grabowski & Węsławski, 2021*). Although the Arctic has a more recent history of glaciation, dated to about 2.5 Ma, than the Antarctic (>20 Ma), the paucity of the temperate and tropical fauna in the Northern Hemisphere is comparable to that of the Southern Hemisphere. Although in the Arctic, the tanaidaceans share some species with the temperate Atlantic and are characterized by a lower level of endemism (*Sieg, 1986a*), tropical/temperate shallow-water apseudomorphs are still underrepresented, including Chondropodinae (*Stępień, Pabis & Błażewicz, 2018*) and plesiomorphic families of Tanaidomorpha.

## CONCLUSIONS

A new tanaidacean genus and species from the subfamily Chondropodinae is described, which can easily be distinguished from other Chondropodinae by having equally long antennular flagella. *Muvi schmallenbergi* is the third Chondropodinae species recorded from Australia after *Bamberus jinigudirus* and *Julmarichardia gutui*. Morphological analysis of the genus *Deidamiapseudes* indicated the transfer of this genus from Chondropodinae (Metapseudidae) to Apseudoidea *incertae sedis*. One of the crucial arguments supporting this reclassification was the absence of a row of setae on the pereopod-1 basis. The absence of Chondropodinae in higher latitudes suggests their extinction during glaciation events and their physiology adapted to tropical/temperate conditions, resulting in their inability to recolonize polar regions.

## ACKNOWLEDGEMENTS

The authors would like to thank Gary Poore for collecting the material and Melanie Mackenzie and Joanne Taylor from the Museum of Victoria, Melbourne Musuem for making the material available for the study. We also thank Andrés G. Morales-Núñez, Richard Heard, and an anonymous reviewer for their constructive comments on the manuscript.

### Funding

The study was funded by the Polish National Science Centre grant 2018/31/B/NZ8/03198. The funders had no role in study design, data collection and analysis, decision to publish, or preparation of the manuscript.

### Grant Disclosures

The following grant information was disclosed by the authors:
Polish National Science Centre: 2018/31/B/NZ8/03198.

### Competing Interests

The authors declare that they have no competing interests.

### Author Contributions

- Piotr Jóźwiak conceived and designed the experiments, performed the experiments, analyzed the data, prepared figures and/or tables, authored or reviewed drafts of the paper, and approved the final draft.
- Magdalena Błażewicz conceived and designed the experiments, performed the experiments, prepared figures and/or tables, and approved the final draft.

### Data Availability

    Specimens are deposited in Melbourne Museum (NMV, Australia):
    Holotype female: MNV J74649; Paratype female: MNV J74648.

## New Species Registration

The following information was supplied regarding the registration of a newly described species:

Publication LSID: urn:lsid:zoobank.org:pub:E516068D-B9FC-4267-BC3C-6C97CF6728C1

*Muvi* gen. nov.: urn:lsid:zoobank.org:act:60F20E13-CC0C-4779-828F-50A561E1BB85

*Muvi schmallenbergi* sp. nov.: urn:lsid:zoobank.org:act:14743564-C6F2-42CE-A181-CE30F5C5A2C2.

## Supplemental Information

Supplemental information for this article can be found online at http://dx.doi.org/10.7717/peerj.11607#supplemental-information.

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
