# Peer review of "Muvi schmallenbergi gen. nov., sp. nov. (Crustacea, Tanaidacea) from the southeast Australian coast, with comments on the distribution and habitat preferences of Chondropodinae"

_PeerJ, doi:10.7717/peerj.11607_

## Round 0.1 · original submission · Major Revisions

I have heard back from three expert reviewers; all consider your work worthy of publication, but all have also recommended many helpful edits and amendments to your work. Please look at these carefully, and also please go over the English very carefully before any resubmission.

·

Basic reporting

The manuscript (ms) describes an undescribed new genus and new species of Tanaidacea from Australia. Which well-deserved to be published. Overall, this manuscript is well-writing and well-structured; however, this ms needs to be amended and improved before its publication, since there are several things, according to my opinion, that needs to be addresses as:

It would be nice to see at least a dorsal drawing of the habitus because the body description was based on a photo.

It would also be nice to have a drawing showing the coxal process/apophysis of pereopod-1.

I would like suggest adding a more detailed description of the pereopods; especially, when the authors are referring to the setation. Because they use very often the phrase “other setation as figured”.

The authors need to review carefully the distribution maps. Because there are several inconsistencies and there are geographical errors at the moment to locate several species.

Experimental design

The paper presents an original study, which will be very useful and will increase the diversity and knowledge of the Tanaidacea fauna from Australia.

Validity of the findings

This is a very useful ms describing a new genus and species of metapseudid. This contribution will increase our knowledge of the Tanaidacea diversity and distribution in Australia waters and worldwide.

Additional comments

Some references are missing and some references included in the text do not match with the reference list or vice-versa.

There are several mistakes/inconsistencies and several points need to be addressed. Please, see attached a revised version of the manuscript (i.e., PDF) with comments, suggestions, and corrections to consider.

Reviewer 2 ·

Basic reporting

The manuscript has been well prepared in its general form, with excellent drawings and competent and thorough taxonomic text and description.

It is well-referenced and the inclusion of a summary of metapseudid sediment "preferences" is a thoughtful and useful contribution.

A key to the genera is useful.

The text (apart from the taxonomic section) is let down by inadequate English grammar and this could be improved, with help from annotations in the returned PDF. It is sometimes too flowery or over-indulgent.

Overall, the hypothesis of a new genus stands up well in the m/s as presented.

Experimental design

The research is well founded and fits within the general aim of defining and enumerating the complex Australian tanaidacean fauna. This is a major expertise of the two authors.

Technically, it is performed to the usual high standards of the authors, combining photographic and graphic methods to interrogate the new taxon. The drawings themselves are of a very high standard and somehow manage to combine good artistry with accurate representation.

A possible addition to the text to improve its clarity and usefulness in interpreting the taxonomy/classification of the new genus would be a Diagnosis of the subfamily Chondropodinae and its relationship/distinction within the family Metapseudidae.

Validity of the findings

This manuscript does provide good evidence for the establishment of a new genus and species within the tanaidacean family Metapseudidae.

I am not convinced by the need for background on the Australian tanaid fauna as a whole in the Introduction, and even here it is not presented in a consistent manner. I think this is covered by other papers and the emphasis should on on the family/subfamily dealt with here.

In a similar way, I think much of the Discussion about tanaid origins (re Arctic/Antarctic and historic patterns) strays too far away from the main thrust of the manuscript, is covered elsewhere and possibly has little relevance to the family discussed. It could be looked at again and revised.

Additional comments

Much of what has been said in the previous review sections does not need to be repeated here, but any revision of the manuscript should concentrate on:

improving the English text

considering the validity and necessity of sections in the Introduction and Discussion

inclusion of a diagnosis for the subfamily (and even family for context).

Otherwise, it is very nice contribution to the inventory of apseudomorphan tanaids and yet another genus to look out for in neighbouring regions such as the eastern IndoPacific/Polynesia and New Zealand.

Annotated reviews are not available for download in order to protect the identity of reviewers who chose to remain anonymous.

Reviewer 3 ·

Basic reporting

.

Experimental design

.

Validity of the findings

.

Additional comments

See attachment.

Annotated reviews are not available for download in order to protect the identity of reviewers who chose to remain anonymous.

---

## Round 0.2 · Minor Revisions

Two of the reviewers from the first round have re-reviewed your work, and scientifically, your work is now generally acceptable (there are a few comments in the attached PDF files). However, the English still needs some brushing up, and I would like you to be thorough and careful in your revision while referring to the attached files. I would like to ask you to have another file check or ask a colleague to do so before your next resubmission.

·

Basic reporting

No commnet

Experimental design

No comment

Validity of the findings

No commnet

Additional comments

The last name of one author cited in the text and Table is misspelled.

There are few mistakes/inconsistencies that needed to be addressed. Please, see attached a detailed revised version of the manuscript (PDF) with comments, suggestions, and corrections to consider.

Reviewer 2 ·

Basic reporting

The revised manuscript is sound according to the criteria set out here.

Experimental design

The revised manuscript is sound according to the criteria set out here.

Validity of the findings

The manuscript provides a good summary of the interesting subfamily Chondropodinae and should provide a clearer basis for other researchers having to deal with the group.

The discussion about historical and current zoogeography remains, unsurprisingly, highly speculative and difficult to avoid circular augments but is still a valid hypothesis.

Additional comments

The authors have made some attempt to pare down the, perhaps, overblown discussion about Antarctic faunas etc in relation to the subfamily Chondropodinae, but the paper can just about accommodate what remains.

The English text still needs some attention and this is indicated in the returned manuscript.

Annotated reviews are not available for download in order to protect the identity of reviewers who chose to remain anonymous.

---

## Round 0.3 · Minor Revisions

I am sorry, but while scientifically this work is ready to be accepted, the editing to the English is not up to international standard, particularly in the Introduction. I would like to ask that you revise the Introduction, hopefully with a colleague or service familiar with biological and taxonomic writing.

---

## Round 0.4 · accepted · Accept

Thank you for your attention to detail. I have read through your responses and revised manuscript, and am happy to move this into production. I look forward to seeing the published version of your work.